# Integrative Stacking Machine Learning Model for Small Cell Lung Cancer Prediction Using Metabolomics Profiling

**DOI:** 10.3390/cancers16244225

**Published:** 2024-12-18

**Authors:** Md. Shaheenur Islam Sumon, Marwan Malluhi, Noushin Anan, Mohannad Natheef AbuHaweeleh, Hubert Krzyslak, Semir Vranic, Muhammad E. H. Chowdhury, Shona Pedersen

**Affiliations:** 1Department of Electrical Engineering, Qatar University, Doha 2713, Qatar; sumon@qu.edu.qa (M.S.I.S.); noushinas71@gmail.com (N.A.); 2College of Medicine, QU Health, Qatar University, Doha 2713, Qatar; mm2308026@qu.edu.qa (M.M.); ma1908120@qu.edu.qa (M.N.A.); svranic@qu.edu.qa (S.V.); 3Department of Clinical Biochemistry, Aalborg University Hospital, 9000 Aalborg, Denmark; h.krzyslak@rn.dk

**Keywords:** SCLC, NSCLC, serum metabolomics, machine learning, stacking ensemble model

## Abstract

This study investigates lung cancer detection by combining metabolomics and advanced machine learning to identify small cell lung cancer (SCLC) with high accuracy. We analyzed 461 serum samples from publicly available data to create a stacking-based ensemble model that can distinguish between SCLC, non-small cell lung cancer (NSCLC), and healthy controls. The model has 85.03% accuracy in multi-class classification and 88.19% accuracy in binary classification (SCLC vs. NSCLC). This innovation relies on sophisticated feature selection techniques to identify significant metabolites, particularly positive ions. SHAP analysis identifies key predictors such as benzoic acid, DL-lactate, and L-arginine, shedding new light on cancer metabolism. This non-invasive approach presents a promising alternative to traditional diagnostic methods, with the potential to transform early lung cancer detection. By combining metabolomics and machine learning, the study paves the way for faster, more accurate, and patient-friendly cancer diagnostics, potentially improving treatment outcomes and survival rates.

## 1. Introduction

Lung cancer remains a leading cause of cancer-related mortality worldwide, with approximately 1.8 million deaths and 2.5 million new cases reported in 2022 [1,2]. Lung cancer is the most common cancer in men and the second most common cancer in women. Non-small cell lung cancer (NSCLC) and small cell lung cancer (SCLC) are the two major lung carcinoma subtypes [3]. The growth rate of SCLC is more rapid than NSCLC and is considered more aggressive [4]. SCLC accounts for about 10–15% of all lung cancers and is known for its strong tendency to metastasize, frequently resulting in widespread disease at the time of diagnosis [5,6]. Biomarker-guided treatment is currently the best method for diagnosing NSCLC and SCLC; however, it requires a biopsy [7]. According to recent literature, the low detection rates for lung cancer at earlier stages are the major contributing factor to its alarming severity [8]. Diagnosis depends on distinct clinical symptoms and pathogenesis, but during this stage, the symptoms are mostly absent, and the pathogenesis is highly complex.

Previous research demonstrated that the metabolism of cancer cells is different from that of normal cells [9]. Metabolites are the byproducts of biological activity, and the change in metabolism, inevitably, causes metabolite concentrations to shift. This changes the unique metabolite profile of the individual [10]. Researchers are exploring metabolites to transform the personalized diagnosis of lung cancer from invasive to non-invasive approaches. Over 150 metabolites have been found to have a connection with lung cancer that reflects its distinct metabolic adaptations [11,12]. Amino acids like serine, glycine, and glutamate are frequently increased, supporting the heightened demands of cancer cell metabolism and proliferation. Alterations in lipid profiles, particularly phosphatidylcholines, sphingomyelins, and fatty acids, have been implicated in lung cancer progression, with lysophosphatidylcholines (LPC) showing notably higher levels in cancerous tissues [13]. Organic acids such as lactic acid and pyruvic acid are elevated due to the Warburg effect, wherein cancer cells favor glycolysis over oxidative phosphorylation, even in oxygen-rich conditions [14]. Increased levels of nucleotides, including uridine and pseudouridine, suggest changes in RNA metabolism associated with lung cancer [15]. Shifts in carnitine levels, especially of long-chain acylcarnitines, point to disruptions in fatty acid oxidation [16]. Additionally, elevated polyamines like spermine and spermidine correlate with increased cell proliferation [17]. Changes in energy metabolites, such as glucose, citrate, and other TCA cycle intermediates, further highlight the altered energy metabolism characteristic of cancer cells [18].

Reaching meaningful statistical interpretations about the measured variables is dependent on the fact that data points are recorded from a broader population. Conventional statistical approaches used in metabolomics concentrate on the establishment of correlations among dependent and independent variables [19]. Machine learning (ML) techniques rely on the use of ad hoc computing algorithms that are either optimized or trained without the need for explicit statistical assumptions [20].

Metabolomics holds the potential to identify the biomarkers of NSCLC and SCLC development. Researchers have developed a prognostic model to predict lung cancer occurrence and patient survival using metabolomics based on nuclear magnetic resonance imaging [21]. In another study, a collection of six metabolites achieved an area under the curve (AUC) value of 0.99 for identifying NSCLC. The six metabolites included are acyl-carnitine C10:1, inosine, L-tryptophan, hypoxanthine, indoleacrylic acid, and LPC [18:2] [22]. Diacetylspermine is known as a unique pre-diagnostic serum biomarker for NSCLC [23]. However, this research only investigated NSCLC detection, and the biomarker of pemetrexed efficacy remains unexplored.

ProGRP and Neuron-specific enolase (NSE) are well-recognized biomarkers of SCLC diagnosis; however, they are found to have low sensitivity and specificity [24,25,26,27,28,29,30,31,32]. For the diagnosis of SCLC, ProGRP had a pooled sensitivity of 72% (95% CI: 47–86%) and a pooled specificity of 93% (95% CI: 95–100%) [33]. Additionally, it exhibited a higher prediction accuracy with a specificity of 72 percent to 99 percent when contrasted with NSE [33]. Some other biomarkers mentioned in previous studies for SCLC diagnosis are lactate dehydrogenase (LDH) and caspase-cleaved cytokeratin 19 (CYFRA21-1), with AUC values of 0.616 and 0.732, respectively [34]. While these are indeed used, they are not specific to SCLC and are more general markers of tumor burden. NSE remains the most widely used serum biomarker for SCLC. In addition, extracellular vesicles (EVs), circulating tumor cells (CTCs), circulating tumor DNA (ctDNA), and exosomes are blood-based tumor components that could serve as alternative ways to evaluate biomarkers for the identification of SCLC [28,35,36,37].

Each tumor hallmark has its limitations and is generally insufficient as a standalone criterion for histological diagnosis or screening [34]. For example, the study of cell-free DNA (cfDNA) has emerged as a promising non-invasive method for tumor detection, as its methylation profiles can be identified at the early stages [4]. However, the results may be biased due to several factors: the methylation levels of CpG sites influence cfDNA, leading to poor sensitivity; cfDNA possesses unstable characteristics that contribute to degradation; and low tumor shedding rates result in very low levels of cfDNA [38].

Machine learning has significant potential in identifying biomarkers for SCLC using metabolomics data. However, due to the limited availability of metabolomics datasets in this field, a considerable research gap still exists. Recently, Shang et al. introduced a large-scale metabolomics dataset specifically for SCLC [4], addressing this gap. Our study utilizes this dataset to conduct a comprehensive machine learning experiment, exploring its potential in biomarker discovery. 

This study advances lung cancer detection by combining metabolomics with machine learning, presenting a stacking-based ensemble model that effectively distinguishes between SCLC and NSCLC while revealing key metabolic signatures through SHAP analysis.

## 2. Methods and Materials

### 2.1. Study Cohort, Sample Collection, and Metabolomics Dataset

We utilized a publicly available LC-MS/MS-based metabolomics dataset as described by Shang et al. [4]. In brief, a total of 501 serum samples were collected from the Shandong Cancer Hospital and Institute between March and November 2020. After excluding samples with incomplete prescription data and suboptimal serum quality, the final dataset included 461 individuals. This cohort consisted of 191 cases of SCLC, 173 cases of NSCLC, and 97 healthy controls. The majority of patients with SCLC were below 65 years of age, with 70.7% of the population being male. Among the NSCLC cases, 91 were diagnosed with lung adenocarcinoma and 82 with squamous cell carcinoma [4]. The comprehensive clinical characteristics are presented in Appendix A. Histopathological analysis was performed to confirm the cancer status of all participants, and blood samples were collected before the initiation of any antitumor therapies. Shang and co-authors obtained serum samples by centrifuging blood at 3000× *g* for 10 min, within 6 h of collection [4]. In our study, we utilized both positive and negative ion mode datasets for metabolomics analysis. Positive mode ionization involves the removal of electrons, generating positively charged ions. This mode is susceptible to detecting metabolites that typically carry a positive charge, such as amino acids, lipids, and small organic molecules like sugars and nucleotides. It is commonly favored for identifying metabolites such as amino acids, neurotransmitters, and lipids. On the other hand, negative mode ionization involves gaining electrons, resulting in negatively charged ions. This mode is more effective for detecting molecules that generally carry a negative charge, such as organic acids, sulfates, and phosphates. Negative ion mode is frequently used for detecting metabolites such as organic acids, fatty acids, and nucleotides. By using both ionization modes, we achieved a more complete representation of the metabolite profiles associated with a wider range of biochemical pathways. The positive mode covers pathways associated with amino acid metabolism, neurotransmitter synthesis, and lipid metabolism, while the negative mode provides insights into pathways involving organic acids, oxidative stress, and phosphate-containing compounds. This dual-mode approach is not only essential to ensure comprehensive metabolic profiling but also allows us to compare our classified metabolites as features in our ensemble model.

### 2.2. Dataset Preprocessing

Dataset preprocessing is the process of converting unprocessed data into a format that ML models can use efficiently and effectively [39]. Preprocessing is essential for addressing issues such as noise, missing values, and unbalanced data, assuring data consistency, and improving model performance. This study focused exclusively on the annotated features from the negative and positive ion metabolomics datasets. The metabolomics analysis was conducted according to the methodology outlined by Shang et al. [4], employing LC-MS/MS in both positive and negative ionization modes to thoroughly identify metabolites. The negative ion dataset contained 158 annotated known features, whereas the positive ion dataset included 152 annotated known features, demonstrating extensive metabolome coverage. Only these annotated features were used in subsequent analyses and model training. For the ML model to achieve optimal training results on our dataset, it was imperative to normalize the input data. To accomplish this, we employed the StandardScaler method to normalize our dataset, which facilitated the robust training of the ML model and facilitated generalized performance [40].

### 2.3. Evaluation Metrics

This study considered a variety of performance evaluation metrics, as relying solely on accuracy is inadequate [41,42]. The performance of the classifiers was assessed using Area Under the Receiver Operating Characteristic Curve (AUC-ROC), in addition to metrics such as Precision, Sensitivity, Specificity, Accuracy, and F1 Score. We employed weighted metrics for each class along with overall precision to account for the differences in the number of instances across classes. Additionally, the AUC score was used as a performance metric. The equations for the evaluation metrics are provided in Appendix A.

### 2.4. Development of Machine Learning and Stacking Ensemble Models

We initially trained more than 10 machine learning models, including tree-based models (CatBoost, RandomForest, ExtraTrees, GradientBoosting, XGBClassifier), ensemble methods (LGBM), linear models (LogisticRegression, ElasticNet), and neural networks (MLPClassifier). This diverse selection was curated to examine a variety of algorithmic approaches, each offering unique strengths for managing complex, high-dimensional data, such as our metabolomics dataset. We focused on tree-based models due to their ability to handle non-linear relationships and interactions between features, which is particularly relevant in metabolomics data where complex biochemical pathways are involved. Ensemble methods were included for their capacity to reduce overfitting and improve generalization. Linear models were selected for their interpretability and efficiency with large feature sets, while neural networks were included for their ability to uncover intricate patterns within the data. After conducting initial evaluations, we proposed a stacking-based ensemble model design. This approach enables us to capture diverse aspects of the metabolomic data that single models may overlook. The final selection of models for the stacking ensemble was guided by their performance metrics and their capacity to provide unique insights to the ensemble. This thoughtful process of model selection and ensemble construction aims to maximize our predictive power while ensuring interpretability, which is essential in a clinical context. The stacking-based approach involves two levels of learners: base learners and meta-learners. The top three performing machine learning classifiers were selected as base learners in the stacking model, while a classifier was employed as the meta-learner in the second stage. This structure ultimately produced the final prediction, aiming to enhance the overall performance of the model.

The proposed stacking ensemble architecture is illustrated in Figure 1.

The essence of this approach involves training a meta-model that adeptly learns to combine the predictions from the base models, thereby enhancing the overall predictive accuracy. Given an input *x* and the predictions from a set of base-level classifiers denoted as *M*, a probability distribution is generated:(1)$$P^{M}\left(x\right)=\left(P^{M}\left(y_{1}|x\right),P^{M}\left(y_{2}|x\right),\ldots ,P^{M}\left(y_{m}|x\right)\right)$$
where ($y_{1}$, $y_{2}$, …, $y_{m}$) represents the possible class values, and $P^{M}$ ($y_{i}$∣*x*) signifies the probability that the instance *x* belongs to the class $y_{i}$, as predicted by classifier *M* [43,44]. In our study, we conducted a multi-class classification analysis for distinguishing between Control (healthy), SCLC, and NSCLC. The CatBoost, RandomForestClassifier, and ExtraTrees models emerged as the top three performers. We utilized their predicted probabilities to train stacking-based meta-models, employing five-fold cross-validation in both the initial and stacking phases. For the binary-class classification (SCLC vs. NSCLC), the MLPClassifier, SVM, and ElasticNet models were identified as the best performers. Their predicted probabilities were leveraged for the initial training, with five-fold cross-validation applied in both phases. An overview of the algorithm is presented in Appendix A.

An overview of the methodology employed in this study is illustrated in Figure 2. This analysis utilized a publicly accessible LC-MS/MS metabolomics dataset that encompassed both positive and negative ion features. These ion features are amalgamated to generate a comprehensive feature set. To extract exploratory insights, univariate analysis techniques, such as clustering and t-SNE, were used; along with these, we also performed PCA. Afterward, the dataset is preprocessed and subjected to feature evaluation to guarantee the quality and relevance of the data. The robustness of the model is improved by employing a five-fold cross-validation strategy. In the subsequent phase, the top three base models are selected after the parallel training of multiple base models. Utilizing the prediction probabilities of these best models, a stacking meta-classifier is trained to consolidate their respective strengths using five-fold cross-validation. For both multi-class and binary classification tasks, this ensemble model that is based on layering generates final predictions. Ultimately, the SHAP analysis is conducted to evaluate the model’s overall impact, analyze the significance of each metabolomics feature, and interpret the model’s predictions, thereby providing information that is pertinent to SCLC.

The study was conducted on a local machine using an Intel(R) Core(TM) i9-10980HK CPU running at 2.40 GHz (up to 3.10 GHz), with 32 GB of RAM and an 8 GB GPU. The Scikit-Learn package, alongside Python 3.10, was used for model training. All models were trained following the specified parameters on this hardware setup to ensure efficient performance. Scikit-learn employs the softmax activation function for multiclass classification problems, while the sigmoid activation function is used for binary classification tasks.

A more detailed overview of this study is illustrated in Appendix A. The t-SNE visualizations for negative and positive ion data are provided in Appendix A, and the PCA component plots for negative and positive ion data are shown in Appendix A, respectively [45]. Additionally, a hierarchical clustering [46,47] heatmap, presented in Appendix A, utilizes metabolomics data to identify key metabolites distinguishing SCLC from NSCLC, underscoring significant metabolic pathways. The parameters of the MLPClassifier for binary classification are shown in Appendix A, and for multiclass classification, the parameters of the MLPClassifier are shown in Appendix A.

## 3. Results

### 3.1. Feature Ranking

Feature ranking is essential in machine learning, as it enhances model performance by recognizing and selecting the most pertinent features, hence eliminating irrelevant or redundant data and improving forecast accuracy. Prevalent methodologies encompass embedded approaches such as XGBoost [48], Random Forest [49], and Extra Trees [50], wherein feature relevance is ascertained during training by evaluating the decrease in loss or impurity. These strategies facilitate the identification of the most meaningful features, enhancing model accuracy and interpretability. This investigation utilizes three advanced machine learning feature selection models: XGBoost, Random Forest, and Extra Trees. In the preliminary analysis, XGBoost demonstrated superior performance for multi-class classification, while Extra Trees outperformed for two-class classification.

For the multi-class classification, optimal results were attained utilizing the top 47 features identified by XGBoost based on feature importance. As illustrated in Figure 3, the most meaningful features comprise 12-Benzene dicarboxylic acid, Phe-Ser, and Phe-Phe, among others. These features significantly enhanced the model’s prediction performance. The importance of each characteristic is presented, with 12-Benzene dicarboxylic acid exhibiting the highest importance, followed by other essential metabolites. From Figure 3, we find that positive ion metabolites are more meaningful than negative ion metabolites. Among the top 10 ranked features, seven are derived from the positive ion mode, indicating their higher contribution to the model’s performance in distinguishing between the classes. This highlights the greater importance of positive ion metabolites in the classification task.

Figure 4 displays the top 48 features ranked by their importance in distinguishing between binary classes: SCLC and NSCLC. The Extra Trees model was utilized to determine feature importance, with each bar representing the relative contribution of a feature to the model’s predictive performance. Key metabolites such as Phthalic acid Mono-2-ethylhexyl Ester, Dioctyl phthalate, and 3-Phosphoserine are among the most influential, with Phthalic acid Mono-2-ethylhexyl Ester showing the highest relative importance. Other meaningful features include Leu-Phe, p-Cresol, and 12-Benzenedicarboxylic acid, which also contribute substantially to the classification task.

### 3.2. Multi-Class Classification

Table 1 represents the detailed results for individual models and stacking models using five-fold cross-validation. The model was trained on a reduced set of the top 48 features, which were selected using the XGBoost feature selection method. The feature selection process was crucial for identifying the most informative features contributing to the ability of the model to distinguish between the three classes. Among the top-performing models, CatBoost exhibited the highest performance across multiple metrics, achieving an accuracy of 84.81%, precision of 84.86%, recall of 84.81%, specificity of 90.08%, and an AUC of 94.82. Other strong performers included SVM (accuracy: 85.03%, precision: 85.05%, recall: 85.03%, specificity: 90.25%, AUC: 92.47) and MLPClassifier (accuracy: 84.82%, precision: 84.87%, recall: 84.81%, specificity: 90.14%, AUC: 94.17). These results reflect the models’ robust classification capabilities, with CatBoost and SVM demonstrating consistently high performance across all key evaluation metrics. Conversely, models like ExtraTrees (accuracy: 80.69%, AUC: 93.45), GradientBoosting (accuracy: 80.47%, AUC: 92.01), and XGB (accuracy: 79.82%, AUC: 92.65) showed somewhat lower accuracy and AUC scores but still offered valuable insight into feature importance and model behavior.

We employed a stacked ensemble approach to further enhance the classification performance, utilizing the prediction probabilities from the top three models. This approach was implemented using five-fold cross-validation to ensure robust evaluation and reliable results. This approach demonstrates a notable performance improvement. The stacking ensemble method significantly improved performance compared to the individual models. For instance, SVM in the stacking ensemble yielded an accuracy of 85.03%, precision of 85.05%, recall of 85.03%, specificity of 90.25%, F1 score of 85.04%, and an AUC of 92.47, showcasing a marginal but notable enhancement over its performance in the initial training. Similarly, the MLPClassifier in the ensemble resulted in accuracy, precision, and recall all surpassing 84.8%, and an AUC of 94.17. These improvements underscore the effectiveness of the ensemble strategy in capturing complementary strengths from each model.

Figure 5 shows that accuracy stabilizes around 80% as the number of top features increases from 5 to 50, with the highest accuracy achieved using the top 47 features, indicating that this subset may offer the optimal balance for model performance. The confusion matrix is included in Appendix A. The confusion matrix shows the performance of a stacking-based ensemble model, with SVM as one of the components, in predicting three classes: Control, SCLC, and NSCLC. The model correctly classified 97 Control cases (top-left), 155 SCLC cases (middle-center), and 140 NSCLC cases (bottom-right). However, it misclassified 36 SCLC cases as NSCLC and 33 NSCLC cases as SCLC, while no Control cases were misclassified as either SCLC or NSCLC. Overall, the matrix demonstrates strong classification performance, particularly in distinguishing Control and SCLC, with some confusion between SCLC and NSCLC.

### 3.3. Binary Classification

In our study of multi-class classification, we observed that the stacking-based SVM model performed effectively for control groups but struggled with SCLC and NCLC. Consequently, we transitioned to a binary classification approach, focusing specifically on the SCLC and NCLC groups. Within this framework, we utilized the Extra Trees feature selection method, which demonstrated superior performance in identifying the most relevant features. Our analysis revealed that the model achieved optimal results using the top 48 features, as illustrated in Figure 4. Table 2 presents the performance metric results for binary classification.

Table 2 highlights the performance metrics for individual ML models in the context of binary classification between SCLC and NSCLC, as well as the performance of stacking models, all evaluated using five-fold cross-validation. These results demonstrate that binary classification yielded improved outcomes compared to the multi-class evaluation. Among the individual models, MLPClassifier, SVM, and ElasticNet performed notably well, with their respective accuracy, precision, recall, specificity, F1 scores, and AUC values listed. Leveraging these three models, we implemented a stacking ensemble approach, training a meta-model based on their combined probabilities. Five-fold cross-validation was also used to ensure robust training and evaluation. This stacking model resulted in further performance improvements, with ExtraTreesClassifier emerging as the top performer, surpassing all other models in terms of classification accuracy and overall evaluation metrics. The binary classification between SCLC and NCLC showed substantial improvement, as evidenced by higher scores across various metrics, particularly in AUC. This indicates that focusing specifically on these two lung cancer subtypes allowed for more refined and accurate predictions.

Figure 6 illustrates the accuracies achieved for the top features, selected by the ExtraTrees feature selection model. The graph highlights the performance improvements observed with increasing feature numbers. The analysis suggests that feature selection plays a critical role in improving classification results, and the ExtraTrees model has been particularly effective in identifying these key features for optimal accuracy.

The confusion matrix for the stacking-based ExtraTrees classifier is presented in Appendix A. The confusion matrices illustrate a clear improvement in classification performance from the multiclass to the binary scenario. In the earlier multiclass model, which classified samples into Control, SCLC, and NSCLC groups, the model performed well for the Control group. However, it exhibited significant confusion between the disease groups, misclassifying 36 SCLC samples as NSCLC and 33 NSCLC samples as SCLC. However, in the binary scenario (SCLC vs NSCLC), the accuracy of the model improved, with only 25 SCLC samples misclassified as NSCLC and 21 NSCLC samples misclassified as SCLC. By removing the Control group, the model was able to focus on distinguishing between the two lung cancer types, resulting in fewer misclassifications and better overall performance.

### 3.4. AUC-ROC Analysis

The AUC-ROC is an essential statistic in machine learning for assessing the efficacy of binary classification algorithms [51]. It measures a model’s capacity to differentiate between positive and negative classes at different threshold levels. An AUC of 1.0 denotes an impeccable classifier, whereas an AUC of 0.5 reflects performance akin to random chance.

Figure 7 illustrates the AUC-ROC curve for the stacking-based SVM classifier in the multi-class classification challenge. The SVM classifier, recognized as the most effective model in Table 1, underwent evaluation using five-fold cross-validation. The three classes included are Control, SCLC, and NSCLC. The ROC curve effectively illustrates the classifier’s capacity to differentiate among the three classes. The weighted average AUC value of 0.92 signifies robust model performance, demonstrating that the stacking-based SVM classifier offers superior discrimination among the three classes. Similarly, the AUC-ROC curve for the stacking-based ExtraTrees classifier is depicted in Figure 8, demonstrating the best binary classification performance, as shown in Table 2. SCLC and NSCLC were distinguished in the binary classification task. The AUC-ROC curve illustrates the trade-off between the True Positive Rate (TPR) and the False Positive Rate (FPR) at differing thresholds for each of the five-fold cross-validation divisions. Weighted average AUC of 0.92 is the consequence of the fold-specific AUC values of 0.84, 0.98, 0.93, 0.94, and 0.92. This consistently high AUC value confirms the ExtraTrees classifier’s robustness and reliability for binary classification between SCLC and NSCLC, indicative of outstanding model discrimination.

### 3.5. SHAP Analysis

SHAP is a versatile tool used to interpret machine learning models by assigning SHAP values to each feature, which helps measure its contribution to the prediction of the model [52]. It provides insights at both the global and local levels, revealing the overall importance of features and their specific influence on individual predictions based on Shapley values from game theory. Positive SHAP values indicate that a feature pushes the prediction higher, while negative values suggest a lowering effect. The ability of the SHAP analysis to work with any machine learning model and its equitable allocation of feature contributions make it a powerful tool for understanding model behavior. We utilized SHAP analysis for both multi-class and binary classification tasks. Figure 9 presents the SHAP summary plot for the multi-class classification model. We employed SHAP analysis to interpret the stacking-based SVM model for a multi-class classification task, including Control, SCLC, and NSCLC. The SHAP summary plot visualizes how different features influence the predictions of the model across these classes. Each feature, such as “3-Phosphoserine” and “Cholesteryl sulfate”, impacts the model output, with SHAP values indicating whether a feature pushes the prediction towards a specific class. The color gradient from blue to red represents feature values, with blue indicating low values and red indicating high values. This color scale further illustrates how varying feature values influence classification decisions for each instance, providing insights into feature importance for distinguishing between Control, SCLC, and NSCLC classes. Figure 10 represents the SHAP summary plot for the binary classification between SCLC and NSCLC. It demonstrates the impact of various metabolic features on the prediction of the model. Each feature listed on the y-axis (e.g., Benzoic acid, L-Arginine, DL-lactate) is associated with SHAP values on the x-axis, indicating how much that feature contributes to pushing the output of the model towards one class or the other. This allows for a quick interpretation of whether a high or low value of a particular feature has a positive or negative effect on the prediction. Features such as Benzoic acid, 1-Oleoyl-sn-glycero-3-phosphocholine, and DL-lactate exhibit considerable spread in their SHAP values, indicating their important role in differentiating between SCLC and NSCLC. The distribution of points for each feature reflects both the range of SHAP values and the variability in feature importance across different samples. From the SHAP summary plot, we can identify the top three features with the most substantial impact on the classification of SCLC: Benzoic acid, DL-lactate, and L-Arginine. These features demonstrate a broad distribution of SHAP values, highlighting their strong influence on the predictions of the model. In particular, high values of Benzoic acid (depicted in red) tend to drive the output of the model positively toward the SCLC class, while lower values of this feature (depicted in blue) have a negative impact. Conversely, features such as Taurine and 1-Oleoyl-sn-glycero-3-phosphocholine have a more pronounced effect on the classification of NSCLC. In the SHAP summary plot, these features show higher SHAP values on the positive side of the axis, suggesting that lower concentrations of L-Lysine and 1-Myristoyl-sn-glycero-3-phosphocholine are more influential in guiding the model toward an NSCLC prediction.

SHAP provides multiple techniques for creating local explanations, which are specific to individual samples, including visualizations such as waterfall plots and force plots. Figure 11A presents a force plot, whereas Figure 11B depicts a waterfall plot. In the waterfall plot (Figure 11B), the x-axis represents the probability of a sample being classified as SCLC, while the y-axis displays the metabolomic features and their corresponding values for that sample. The plot begins with the expected value of the model on the x-axis, noted as E[f(X)] = 0.8. This ‘base’ value of 0.8 indicates the average prediction probability across the test set. Subsequently, the plot illustrates how metabolomic features impact the output of the model. Positive contributions (depicted in red) and negative contributions (depicted in blue) modify the expected value to reach the final model output of f(x) = 0.017. Positive SHAP values enhance the probability of the sample being classified as SCLC, while negative SHAP values reduce this likelihood. Additionally, the force plot employs an additive force layout to display the SHAP values of each feature for a specific sample.

## 4. Discussion

SCLC is a fast-growing cancer linked to smoking that often spreads before clinical diagnosis. This cancer has high relapse rates despite chemotherapy response, lowering long-term survival [53]. However, NSCLC progresses slowly and responds better to early intervention [54]. SCLC is aggressive because of its neuroendocrine markers like synaptophysin, chromogranin A, and CD56, which promote cell growth and spread [55]. Our study illustrates the potential of combining metabolomics data with sophisticated machine learning techniques to improve the classification of SCLC and NSCLC. The research employed a publicly accessible LC-MS/MS-based metabolomics dataset that included 191 SCLC cases, 173 NSCLC cases, and 97 healthy controls.

A stacking-based ensemble machine learning approach was employed, which effectively leveraged the strengths of individual models, resulting in improved accuracy, precision, and overall classification performance. For multi-class classification, the SVM within the ensemble achieved an accuracy of 85.03% and an AUC of 92.47. To further enhance performance and reduce ambiguity between SCLC and NSCLC, a binary classification was subsequently conducted, focusing specifically on distinguishing between these two classes. This refinement simplified the classification task, strengthened the model’s robustness, and enabled a more targeted differentiation between SCLC and NSCLC in later stages. The ExtraTreesClassifier within the ensemble attained an accuracy of 88.19% and an AUC of 92.65%. The study identified key metabolites such as benzoic acid and DL-lactate as crucial in distinguishing between SCLC and NSCLC. This approach shows promise for developing non-invasive diagnostic tools that could lead to earlier detection and improved outcomes for lung cancer patients, addressing the limitations of conventional biopsy methods.

The analysis revealed that positive ion metabolites contributed significantly more to the model than negative ion metabolites, particularly in multi-class classification. This observation could be attributed to a variety of biological factors. Modifications to amino acids and their derivatives are critical targets for altered cancer metabolism [13]. Furthermore, the synthesis of lipids such as phosphatidylcholines and sphingomyelins is frequently upregulated to support rapid cell division and membrane production [56]. This mode may also preferentially ionize metabolites involved in energy production, such as carnitine and acylcarnitine, which are frequently altered in cancer cells due to shifts in energy metabolism [16]. These findings may guide future metabolomic studies in lung cancer to prioritize positive ion mode analyses, thereby improving the sensitivity and specificity of metabolomic biomarkers for SCLC.

Furthermore, we employed hierarchical clustering to identify co-regulated metabolites that may share biological pathways, aiding in identifying potential biomarkers and understanding disease mechanisms. SHAP analysis provided deep insights into the prediction of the model, identifying key metabolites like benzoic acid, DL-lactate, and L-arginine as significant discriminators between SCLC and NSCLC, which could guide the development of targeted therapies or diagnostic tools. These findings align with and expand upon established hallmarks of cancer metabolism. The identification of DL-lactate as a significant metabolite supports the Warburg effect, a well-known feature of cancer metabolism that is characterized by increased glycolysis and lactate production, even in the presence of oxygen [13,14]. Additionally, the identification of L-arginine as a significant metabolite in our study aligns with its known role in cancer progression. L-arginine contributes to tumor growth through two main pathways: polyamine production and nitric oxide synthesis [17]. The identification of 12-Benzenedicarboxylic acid as a significant feature in our study points to potential alterations in lipid metabolism. Recent research has shown that cancer cells often exhibit reprogrammed lipid metabolism to support rapid proliferation [56,57]. Our findings suggest that this reprogramming may differ between SCLC and NSCLC. The presence of benzoic acid in our significant metabolites list may indicate alterations in cellular redox balance [58]. Cancer cells often have heightened oxidative stress and altered antioxidant mechanisms [58,59]. The differential levels of benzoic acid between SCLC and NSCLC could reflect varying strategies for managing oxidative stress in these cancer types. Consequently, the discoveries concerning these metabolites have the potential to function as biomarkers and provide a deeper understanding of the metabolic reprogramming that is responsible for various lung cancer subtypes.

Although the data used for model development were derived from a publicly available case–control study with approximately balanced classes, it is important to consider the potential impact of class imbalance when applying the model to populations with naturally occurring class distributions. In real-world scenarios, where the prevalence of certain classes may vary significantly, techniques such as the Synthetic Minority Over-sampling Technique (SMOTE) [60] can be employed to address this challenge and improve model performance. The dataset in our study comprises tabular 1D metabolomics data, in which each row corresponds to a sample and the columns represent specific metabolite features. The classical machine learning models [61] were chosen due to the structured and tabular nature of the data, as they are well-suited for feature-driven datasets. Particularly when employing limited sample sizes, these models provide interpretability, effectiveness, and computational efficiency. Although deep learning models are typically more effective with high-dimensional data and larger sample sizes, recent developments, such as Transformer-based deep learning models [62,63], have shown potential applications for tabular data. Our objective in future research is to investigate these sophisticated deep learning methodologies in order to improve the predictive and analytical capabilities of metabolomics data.

Despite the promising results, the study has several limitations. First, the small dataset may limit the generalizability of the model to larger and more diverse populations. Our study develops a metabolomics-based diagnostic model using 461 people (191 with SCLC, 173 with NSCLC, and 97 controls). To contextualize this sample size, comparing it with other lung cancer metabolomics studies is important. For instance, Shang et al. employed the same publicly available LC-MS/MS-based metabolomics dataset, splitting their data into a training cohort of 323 and a testing cohort of 138 to validate a deep learning model [4]. In contrast, our study employed a training cohort of 369 samples and a testing cohort of 92 samples to evaluate our ensemble approach using five-fold cross-validation. Similarly, Wikoff et al. [23] examined a different dataset, which included 108 NSCLC cases and 216 controls. Although our sample sizes are comparable to those of these studies, particularly in the case of SCLC, they are lacking in comparison to larger-scale NSCLC studies, such as those conducted by Mathe et al. (2014), which included 1005 patients [64]. The small control group and the focus on serum samples, which might not fully capture the tumor microenvironment’s metabolic complexity, further limit generalizability. Additionally, our metabolomics dataset is significantly smaller than those in lung cancer genomic studies, which employ thousands of samples. Campbell et al. (2016) analyzed 1144 lung cancer whole-genome sequencing data points [65]. This comparison demonstrates that to enhance the statistical power of genomic studies and the discovery of biomarkers in lung cancer research, it is necessary to have larger metabolomics datasets. Second, serum samples were used for metabolomics, which may not capture the full complexity of tumor microenvironment metabolic changes. Serum analysis illuminates systemic metabolism, but it often misses tissue-specific metabolic processes crucial to tumor progression. The computational complexity of the stacking ensemble model enhanced performance; however, it may restrict clinical decision-making in fast-paced environments. In summary, the SHAP analysis improved the interpretability of the ML model; however, further clinical validation is necessary to confirm the biomarker relevance of the identified metabolites.

Moreover, to validate and potentially refine the model in future research, it is recommended that larger multi-institutional datasets be employed. Incorporating additional omics data, such as genomics, lipidomics, transcriptomics, and proteomics, could achieve a more holistic view of the molecular landscape in lung cancer [66,67]. Combining our metabolomics data with this multi-omics approach could elucidate synergistic effects and regulatory mechanisms that are not apparent from metabolomics alone, thereby facilitating a more comprehensive understanding of cancer pathogenesis. A recent proteomic profiling systematic review identified several proteins that could potentially play a role in SCLC pathogenesis [6]. Therefore, further investigation of the biological mechanisms and interplay amongst different omics data could guide the development of targeted therapies or diagnostic tools for SCLC.

Our findings emphasize the potential for developing metabolomics-based diagnostic tools for early SCLC detection and personalized treatment strategies. Integrating machine learning algorithms, such as the stacking ensemble model used in this study, could improve diagnostic accuracy and guide treatment decisions. It will be critical to work with clinical experts to establish appropriate thresholds and incorporate predictive models into clinical workflows. Furthermore, our proposed model could be a valuable component of these diagnostic tools, which should also include human expert feedback, additional biomarkers, and complementary predictive models. Future research should investigate the potential of a higher-level ensemble model that combines our metabolomics-based machine learning approach with existing lung cancer screening protocols, such as low-dose CT scans. This integrated system could further automate and enhance the medical decision-making process by providing an additional layer of metabolic information, potentially reducing false positives and unnecessary invasive procedures.

## 5. Conclusions

In conclusion, this study highlights the significant potential of integrating metabolomics data with advanced machine-learning techniques to improve the classification of SCLC and NSCLC. By employing a stacking-based ensemble approach, we achieved enhanced accuracy and precision in distinguishing between cancer types. SHAP analysis revealed key metabolites such as benzoic acid and DL-lactate as critical for differentiation. The clinical implications of the model are noteworthy; it could aid in risk stratification, allowing for personalized management strategies based on metabolic profiles, and potentially informing treatment decisions by identifying potential drug targets. Additionally, its integration into existing screening processes could reduce healthcare costs by minimizing unnecessary invasive procedures. Future research should focus on longitudinal studies to track metabolic profile changes over time and further validate the effectiveness of the model in clinical settings. Overall, our findings accentuate the promise of combining metabolomics with machine learning to develop non-invasive diagnostic tools that could lead to earlier detection and improved outcomes for lung cancer patients.

## Figures and Tables

**Figure 1 cancers-16-04225-f001:**
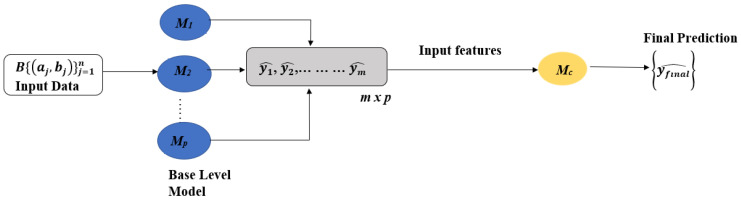
Proposed Stacking Ensemble Model.

**Figure 2 cancers-16-04225-f002:**
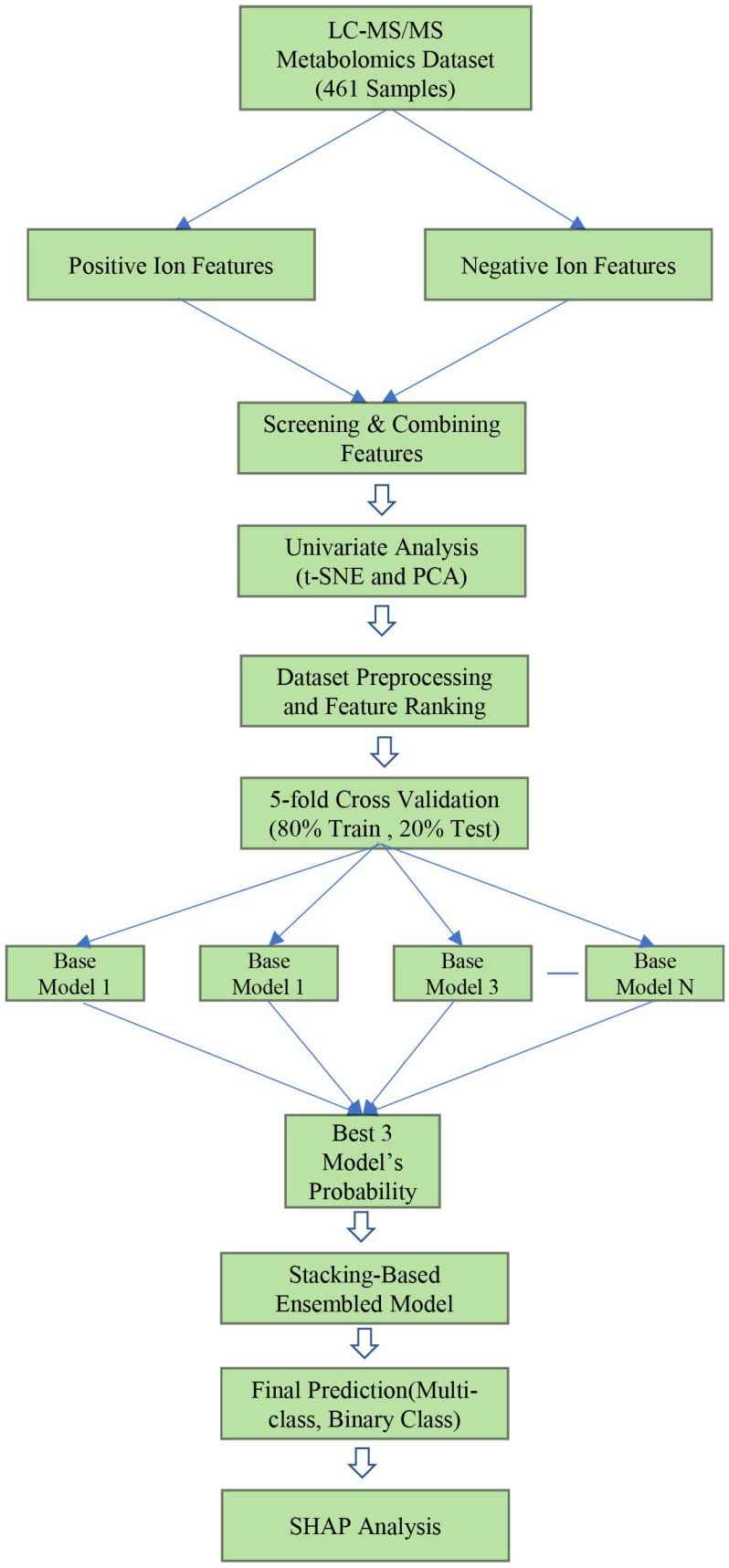
Overview of the methodology employed in this study.

**Figure 3 cancers-16-04225-f003:**
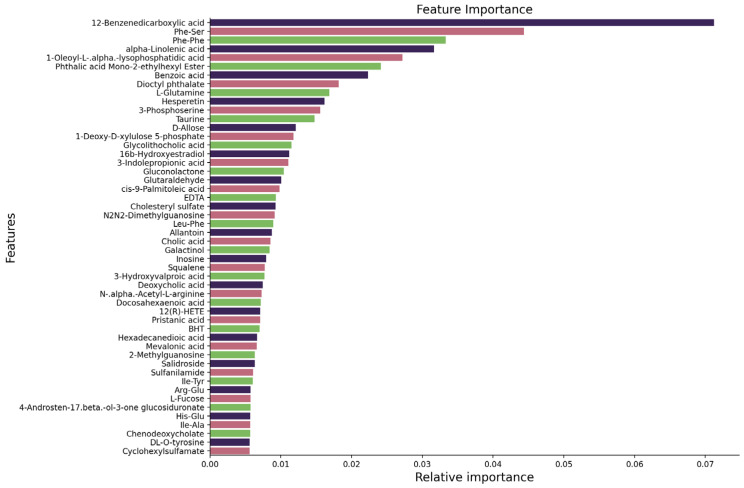
Features ranked using the XGBoost feature selection algorithm for multi-class classification.

**Figure 4 cancers-16-04225-f004:**
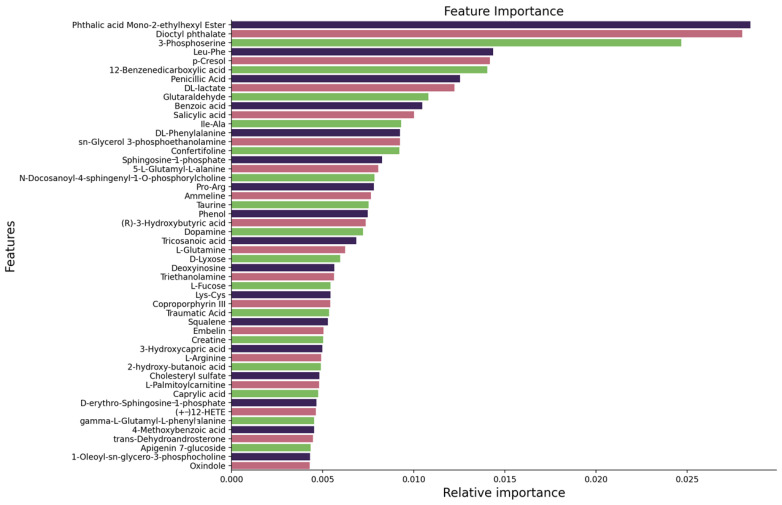
Features ranked using the Extra Trees feature selection algorithm for binary classification.

**Figure 5 cancers-16-04225-f005:**
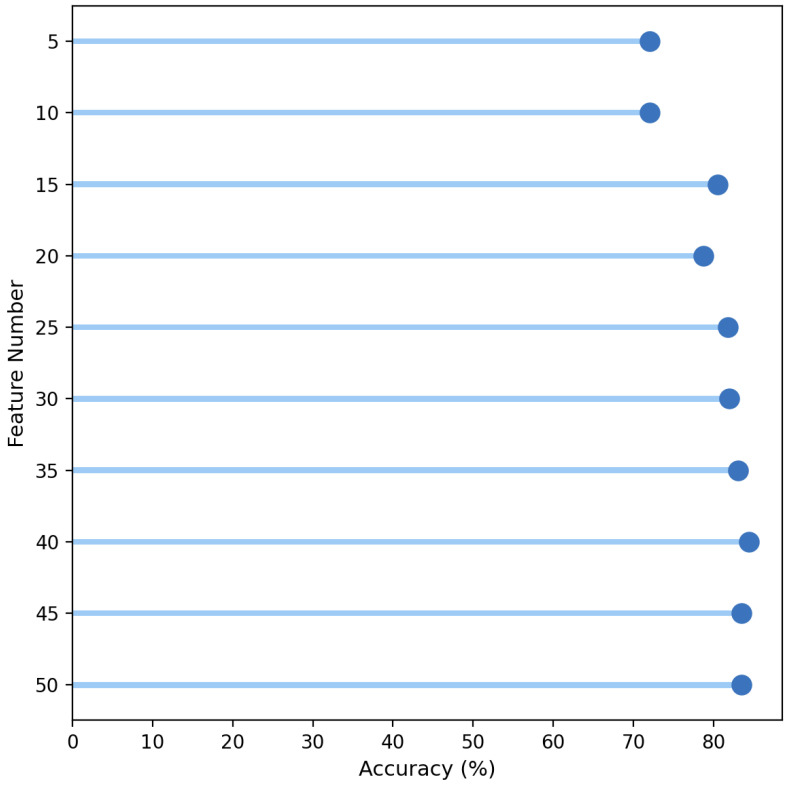
Accuracy of top features for multiclass classification.

**Figure 6 cancers-16-04225-f006:**
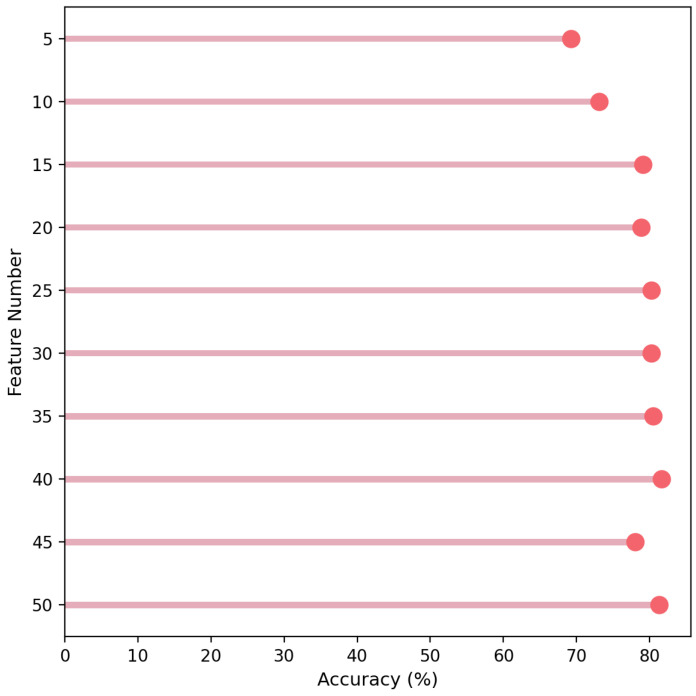
Top Feature Accuracy for Binary Classifications.

**Figure 7 cancers-16-04225-f007:**
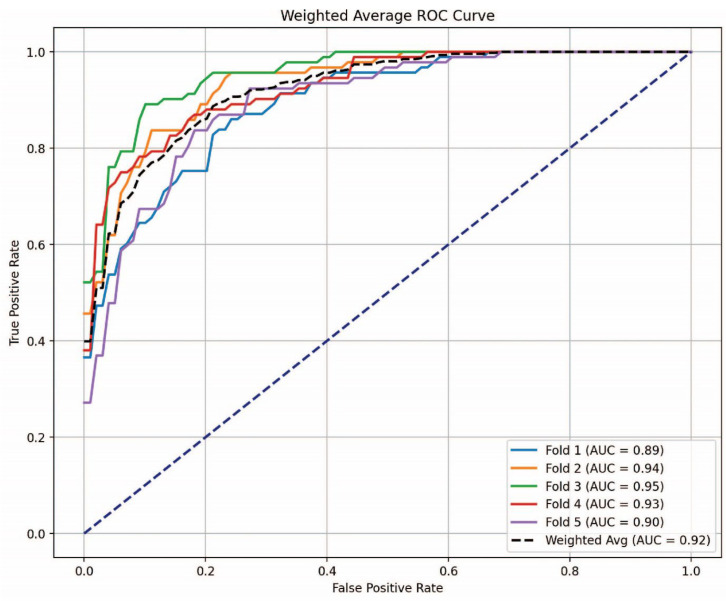
AUC-ROC curve for the stacking-based SVM classifier in multi-class classification.

**Figure 8 cancers-16-04225-f008:**
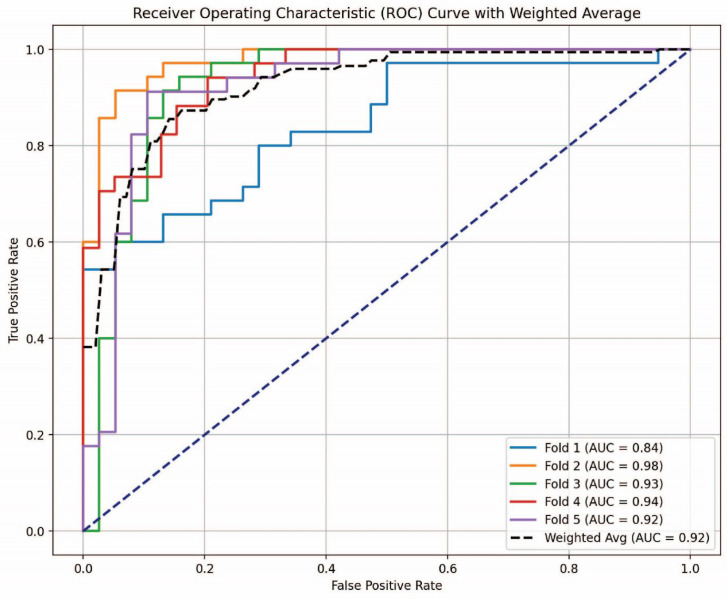
AUC-ROC curve for the stacking-based ExtraTrees classifier in binary classification.

**Figure 9 cancers-16-04225-f009:**
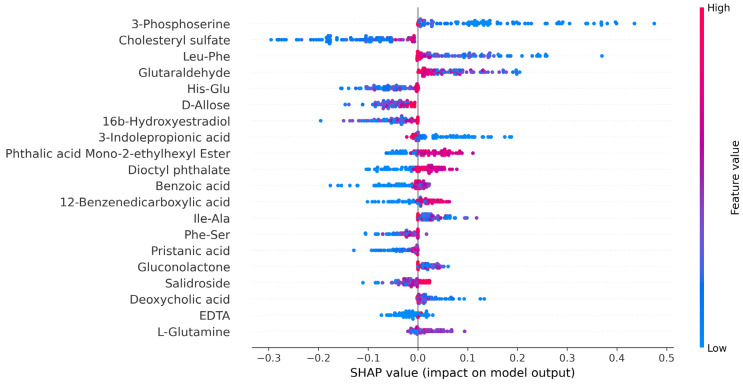
SHAP summary plot for the multi-class classification model.

**Figure 10 cancers-16-04225-f010:**
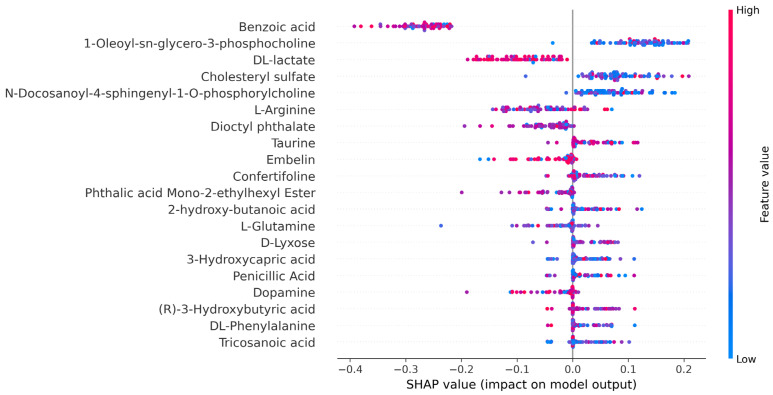
SHAP summary plot for the binary classification model.

**Figure 11 cancers-16-04225-f011:**
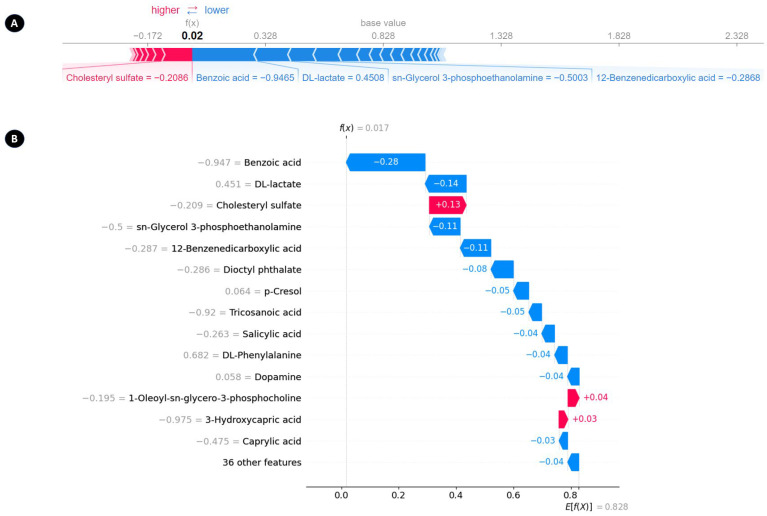
Local explanations of a representative sample are shown in two forms: (**A**) a force plot illustrating an SCLC prediction and (**B**) a waterfall plot displaying the same prediction.

**Table 1 cancers-16-04225-t001:** Top-Performing ML Models and Stacked Ensemble ML Models for Multi-Class Using Five-Fold Cross-Validation.

Initial Results							Stacking Ensemble Results						
Models	A	P	R	S	F1	AUC	Models	A	P	R	S	F1	AUC
**CatBoost**	**84.81**	**84.86**	**84.81**	**90.08**	**84.82**	**94.82**	**SVM**	**85.03**	**85.05**	**85.03**	**90.25**	**85.04**	**92.47**
**RandomForest**	**84.59**	**84.55**	**84.59**	**90.04**	**84.57**	**94.64**	MLPClassifier	84.82	84.87	84.81	90.14	84.82	94.17
**ExtraTrees**	**80.69**	**80.71**	**80.69**	**87.44**	**80.64**	**93.45**	LDA	84.82	84.84	84.81	90.11	84.82	94.06
GradientBoosting	80.47	80.76	80.47	87.23	80.54	92.01	LogisticRegression	84.6	84.6	84.6	89.94	84.6	94.29
XGB	79.82	79.57	79.82	87.67	79.66	92.65	ExtraTrees	84.6	84.6	84.6	89.94	84.6	94.25
LGBM	79.17	79.34	79.17	86.4	79.24	92.37	ElasticNet	84.6	84.6	84.6	89.94	84.6	94.29
LogisticRegression	78.09	78.22	78.09	85.68	78.1	91.58	XGBClassifier	84.16	84.17	84.17	89.65	84.17	93.43
SVM	77	77.35	77	84.97	77.08	89.26	LGBM	83.51	83.51	83.51	89.17	83.5	93.8
MLPClassifier	78.741	78.832	78.741	86.202	78.759	91.37	RandomForest	83.08	83.11	83.08	88.98	83.08	93.4
ElasticNet	76.13	76.06	76.13	84.67	76.06	91.18	CatBoost	81.34	81.37	81.34	87.84	81.35	93.67

Bold values indicate the best performance across all models.

**Table 2 cancers-16-04225-t002:** Top-Performing ML Models and Stacked Ensemble ML Models for Binary Class Using Five-Fold Cross-Validation.

Initial Results							Stacking Ensemble Results						
Models	A	P	R	S	F1	AUC	Models	A	P	R	S	F1	AUC
**MLPClassifier**	**85.43**	**85.43**	**85.43**	**85.43**	**85.43**	**92.04**	**ExtraTreesClassifier**	**88.19**	**88.27**	**88.19**	**88.32**	**88.19**	**92.65**
**SVM**	**84.61**	**84.67**	**84.61**	**84.61**	**84.62**	**93.06**	XGBClassifier	87.36	87.4	87.36	87.41	87.37	92.22
**ElasticNet**	**84.61**	**84.62**	**84.61**	**84.61**	**84.61**	**93.19**	LogisticRegression	86.81	86.82	86.81	86.81	86.82	92.65
RandomForest	83.51	83.61	83.51	83.51	83.52	89.71	CatBoost	86.81	86.88	86.81	86.91	86.82	92.76
LogisticRegression	83.51	83.52	83.51	83.51	83.52	92.49	ElasticNet	86.81	86.82	86.81	86.81	86.82	92.72
LinearDiscriminantAnalysis	83.24	83.36	83.24	83.24	83.25	91.47	LinearDiscriminantAnalysis	86.54	86.58	86.53	86.61	86.54	92.3
CatBoost	82.69	82.86	82.69	82.69	82.7	90.05	SVM	86.54	86.56	86.54	86.56	86.54	92.27
ExtraTrees	81.86	81.86	81.86	81.86	81.85	90.14	LGBM	86.54	86.63	86.54	86.17	86.5	92
AdaBoostClassifier	79.94	80.02	79.94	79.94	79.95	85.92	MLPClassifier	85.99	86.16	85.99	86.22	85.99	92.25
LGBM	76.64	76.86	76.64	76.64	76.65	82.72	RandomForest	85.99	86.03	85.99	86.06	85.99	91.62

Bold values indicate the best performance across all models.

## Data Availability

Shang et al. [4] have provided the dataset that this investigation employs. Therefore, the authors of this article were not involved in the data collection procedure.

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
