# Peer review of "Integrative Stacking Machine Learning Model for Small Cell Lung Cancer Prediction Using Metabolomics Profiling"

_cancers, 2024, doi:10.3390/cancers16244225_

Round 1

Reviewer 1 Report

Comments and Suggestions for Authors

1.      The performance and evaluation metrics employed are adequate. However, given the objective of the model, I’d also include the positive predictive value and the negative predictive value.

2.      Exploratory analysis was performed using clustering and t-SNE. In my opinion, t-SNE is useful for exploratory analysis, but it should be complemented with PCA.

3.      The authors state that they used 5-fold cross validation to train and evaluate the different base models (Figure 2). That is fine. However, there is no mention of how the final predictions from the stacking-based ensemble model were evaluated. When reviewing supplementary figures it seems that they also used 5-fold cross validation, but it should be more clear in the text

4.      Regarding the ROC curves, the five different curves for the different folds show significant variance. This points to considerable uncertainty in the determination of the real AUC value. The authors should provide an estimate of that uncertainty for each of the reported AUC values.

5.      In the results section, I would avoid using the term “significant” for the most important variables so that it is not misleading (i.e., statistically significant).

6.      The data used for the development of the model comes from a publicly available dataset from a case-control study. That means that the number of classes on each category is approximately balanced. The authors should comment on the possible impact of class imbalance when using this model to predict on patients randomly sampled from the population. 

Author Response

Dear Reviewer,  

Thank you very much for taking the time to review our manuscript. Please find the detailed responses below and the corresponding revisions in the attached filesWe have also included the revised manuscript.   

Reviewer 2 Report

Comments and Suggestions for Authors

This study introduces an XAI approach using metabolomics data to classify SCLC and non-small cell lung cancer (NSCLC). Significant metabolites were identified, with positive ions being more relevant. The stacking ensemble model achieved 85.03% accuracy and 92.47 AUC for multi-class classification (control, SCLC, NSCLC) using Support Vector Machine (SVM). For binary classification (SCLC vs. NSCLC), ExtraTreesClassifier reached 88.19% accuracy and 92.65 AUC. Key metabolites like benzoic acid, DL-lactate, and L-arginine were significant predictors. The model is interesting and may gain many interests. However, I have the following comments:

-How the model avoids overfitting?

- I suggest to include the AUROC curve for the models to the manuscript.

-if possible, to enhance Figure 4 (the quality by generating vector based figure).

Author Response

Dear Reviewer,

Reviewer 3 Report

Comments and Suggestions for Authors

The authors state that It is important to recognize that preprocessing plays a crucial role in preparing the dataset for analysis; however, the specifics of the preprocessing steps undertaken are not clearly detailed. Additionally, while there is significant emphasis on the annotated features within the dataset, the discussion lacks concrete examples. Without illustrative references, readers may find it challenging to understand precisely what the authors mean by these features and how they were annotated. Providing specific examples of the preprocessing methods and the annotated features would greatly enhance clarity and comprehension for the audience.

The authors report that they trained a total of ten machine learning models, beginning with a series of traditional models before progressing to a neural network model. While traditional models tend to have fewer hyperparameters, making them easier to interpret and map, the introduction of a neural network necessitates a more nuanced discussion. Neural networks are often likened to “black boxes” due to their complex and opaque nature, which can obscure the rationale behind their predictions. To enhance clarity and facilitate understanding, the authors should specify the type of neural network utilized—such as a feedforward network, convolutional neural network (CNN), or recurrent neural network (RNN)—and provide details about the specific hyperparameters involved in training this model. It is important to recognize that there are substantial differences among various machine learning models, and grouping them under a broad category fails to capture the unique characteristics and performance implications associated with different types. Therefore, a detailed comparison and explanation would help elucidate the methodology and the reasons for choosing each model in the study.

The authors mention both multi-class and binary classifications, which could lead to some assumptions by readers regarding their meanings. A reader might interpret the multi-class classification as representing different types of cancer, while the binary classification could suggest a simple yes or no outcome regarding the presence of cancer. However, it is possible that this interpretation does not align with the authors’ intended message. To avoid any confusion and clarify their intentions, it is essential for the authors to provide a clear explanation of what each classification represents in the context of their work.

Author Response

Dear Reviewer,

Round 2

Reviewer 1 Report

Comments and Suggestions for Authors

The authors have correctly addressed all the raised issues. I don't have any more comments.

Author Response

Dear Reviewer,

Thank you so much for your feedback and support!

Reviewer 3 Report

Comments and Suggestions for Authors

Based on the feedback received, the authors were provided with readily preprocessed data. Subsequently, they assessed for missing data and normalized the dataset. While data normalization is not a prerequisite when utilizing traditional models, its significance increases when employing a multilayer perceptron. In such cases, it is essential not only to normalize the data but also to implement one-hot encoding in the final classification layer.

The response to the second comment appears to offer generic hyperparameter tuning options provided by a library. It’s important to note that while ReLU is an effective squashing function, it is not a good choice for the classification layer. The authors did not specify which activation function was used in the final classification layer. Additionally, the supplementary table suggests that the authors relied on the default values for the multilayer perceptron (MLP) without fully understanding the capabilities of this neural network. It is encouraged to remove the MLP from the options and keep only with traditional models as it is not a fair comparison between the selected algorithms.  

A large number of features in a dataset can significantly increase the risk of overfitting. This phenomenon occurs when a model learns the noise and random fluctuations in the training data rather than identifying the underlying patterns that are truly representative of the domain. Many of these features may be irrelevant or redundant, contributing little to the model's overall predictive power and potentially confusing the learning process. There appears to be an increased number of features that have less than a 1% impact on model predictions. These may be good candidates for recursive feature elimination.

To enhance the model's ability to generalize to unseen data, it is essential to employ feature reduction techniques. These techniques aim to identify and retain only the most informative features while eliminating those that do not provide meaningful insight. By reducing the dimensionality of the dataset, not only does the model become simpler and more interpretable, but it also minimizes the chances of overfitting. Common methods for feature reduction include techniques such as Principal Component Analysis (PCA), Recursive Feature Elimination (RFE), and feature selection algorithms that evaluate the importance of each feature based on its contribution to the predictive performance. Employing these strategies improves the robustness of the model, leading to better performance on new, unseen data.

Author Response

Dear Reviewer,
